# Prediction of Clavien Dindo Classification ≥ Grade III Complications After Epithelial Ovarian Cancer Surgery Using Machine Learning Methods

**DOI:** 10.3390/medicina61040695

**Published:** 2025-04-10

**Authors:** Aysun Alci, Fatih Ikiz, Necim Yalcin, Mustafa Gokkaya, Gulsum Ekin Sari, Isin Ureyen, Tayfun Toptas

**Affiliations:** 1Department of Gynecologic Oncology, Health Sciences University Antalya Training and Research Hospital, Antalya 07100, Turkey; zinaryal@gmail.com (N.Y.); mugokkaya@gmail.com (M.G.); drekingulsum@gmail.com (G.E.S.); isin.ureyen@gmail.com (I.U.); drttoptas@gmail.com (T.T.); 2Department of Emergency Medicine, Health Sciences University Beyhekim Training and Research Hospital, Konya 42060, Turkey; sultanmehmet01@hotmail.com

**Keywords:** artificial intelligence, Bayesian network model, Clavien–Dindo classification, diagnosis, machine learning models, ovarian cancer surgery

## Abstract

*Background and Objectives:* Ovarian cancer surgery requires multiple radical resections with a high risk of complications. The aim of this single-centre, retrospective study was to determine the best method for predicting Clavien–Dindo grade ≥ III complications using machine learning techniques. *Material and Methods*: The study included 179 patients who underwent surgery at the gynaecological oncology department of Antalya Training and Research Hospital between January 2015 and December 2020. The data were randomly split into training set n = 134 (75%) and test set n = 45 (25%). We used 49 predictors to develop the best algorithm. Mean absolute error, root mean squared error, correlation coefficients, Mathew’s correlation coefficient, and F1 score were used to determine the best performing algorithm. Cohens’ kappa value was evaluated to analyse the consistency of the model with real data. The relationship between these predicted values and the actual values were then summarised using a confusion matrix. True positive (TP) rate, False positive (FP) rate, precision, recall, and Area under the curve (AUC) values were evaluated to demonstrate clinical usability and classification skills. *Results*: 139 patients (77.65%) had no morbidity or grade I-II CDC morbidity, while 40 patients (22.35%) had grade III or higher CDC morbidity. BayesNet was found to be the most effective prediction model. No dominant parameter was observed in the Bayesian net importance matrix plot. The true positive (TP) rate was 76%, false positive (FP) rate was 15.6%, recall rate (sensitivity) was 76.9%, and overall accuracy was 82.2% A receiver operating characteristic (ROC) analysis was performed to estimate CDC grade ≥ III. AUC was 0.863 with a statistical significance of *p* < 0.001, indicating a high degree of accuracy. *Conclusions*: The Bayesian network model achieved the highest accuracy compared to all other models in predicting CDC Grade ≥ III complications following epithelial ovarian cancer surgery.

## 1. Introduction

Ovarian cancer is the fifth most common cause of death in women and has the poorest prognosis of all gynaecological cancers [1]. An important reason for the poor outcome of the disease is that most cases are diagnosed at stage III [2]. The current gold standard of care for this gynaecological malignancy is a combination of surgery and platinum-based chemotherapy. However, the role of PARP (poly (ADP-ribose) polymerase inhibitors), especially in BRCA-mutation-positive ovarian cancer or ovarian cancer with a deficiency in the homologous recombination repair pathway, has become increasingly important in the last decade, and successful results have been achieved [3]. The most significant predictors of outcome in patients undergoing cytoreductive surgery for ovarian cancer are the presence of residual tumour and the duration of platinum resistance [4,5]. A meta-analysis has shown that for every 10% increase in the optimal cytoreduction rate, the median survival time is increased by 5.5% [6].

In particular, upper abdominal surgery (diaphragmatic peritonectomy, splenectomy, distal pancreatectomy, etc.), colorectal resections and anastomoses, genitourinary resections, and increased radicality are associated with higher perioperative morbidity and mortality [7,8,9]. The Clavien–Dindo classification system (2004) was developed to categorise surgical complications according to the severity and outcome of the treatment required. It is based on the work of Professor Michel Clavien and his team. This classification serves as a standard reference for both research and clinical studies and also assists surgeons in the assessment and management of complications. Complications classified as grade 3 and above are defined as serious complications requiring surgical, radiological or endoscopic intervention and are a standard reference in the literature [7]. We used this classification system in our study.

Complications after surgery are associated with reduced survival if chemotherapy is started more than 35–42 days after surgery [8,9]. It is important to note that a careful balance is required between the survival benefits of surgical interventions and the potential morbidity risks associated with such procedures. In order to achieve this, there is increasing interest in the development of predictive models that focus on the prediction of post-surgical complications. These models are based on factors such as various patient demographics, disease characteristics and surgical techniques, and they allow for individualisation of treatment protocols by performing patient-specific risk assessments. Developed predictive models aim to reduce postoperative morbidity rates and improve patient safety by optimising perioperative patient management. These approaches require multidisciplinary collaboration and the use of emerging bioinformatics tools [10].

Although scoring systems such as the College of Surgeons Surgical Risk Calculator (ACS-SRC), the American Society of Anaesthesiologists (ASA) score, and the Operative Severity Score for the Enumeration of Mortality and Morbidity (POSSUM) provide important benefits in supporting clinical decision-making, they have some limitations [11,12,13]. Existing risk assessment models are known to be of two main types: Cox proportional hazards regression analysis and logistic regression models. These models are based on variables that have statistically significant effects on the results but may exclude subtle but important factors that may influence the results. Furthermore, regression analyses cannot be applied in studies with too many variables (e.g., LR analysis requires 10 positive outcomes for each variable (10 EPV rule), and certain assumptions, such as Box–Tidwell, need to be met). Since these models assume a linear correlation, they are limited in their ability to account for ranges of variables that extend to extremes and may miss other important modulators of the outcome. In contrast, machine learning in high-dimensional models is emerging as a highly effective and feasible approach in contrast to traditional statistical methods [14,15].

The first artificially intelligent machine with learning capabilities was proposed in the 1950s [16]. Most machine learning (ML) algorithms use different learning approaches, one of which is supervised learning. This technique learns and generalises the features of the training set and makes predictions using features from the test dataset [17]. Supervised ML approaches have been widely used in disease prediction and patient outcome [18]. ML algorithms allow a greater number of clinical variables to be evaluated and can help identify poor predictors or interactions between variables that can improve prediction accuracy. Furthermore, ML is suitable for incremental learning, whereby the model can be improved as new data are incorporated [14,19].

ML is a game-changer in oncology and is ushering in a new era of precision medicine. Based on its ability to help scientists decipher complex patterns in large datasets, ML promises to redefine the paradigms of cancer detection, diagnosis, prognosis and treatment optimisation. The key feature of ML applications is that they use thousands of combinations and complex algorithms to reach a conclusion that is beyond the capacity of traditional statistical methods and human ability. Once the best algorithms for predictive modelling have been identified, these algorithms can be integrated into applications and desktop tools, allowing predictions to be made on the same data [20].

To the best of our knowledge, our study is the first study in the literature and its aim is to determine the ML model that most accurately predicts Clavien–Dindo grade ≥ III morbidity after ovarian cancer surgery and to investigate how ML models can be applied in clinical practice.

## 2. Materials and Methods

The current study was carried out in accordance with the Declaration of Helsinki and approved by the Ethics Committee of Antalya Training and Research Hospital, Antalya, Turkey (approval no: No:6/25/dated 9 May 2024). The data were collected from consecutive patients aged 18 years and older who underwent surgery for epithelial ovarian cancer between January 2015 and December 2020. Patients operated for other gynaecological malignancies, patients with benign histology, and patients under 18 years of age were excluded. The clinicopathological characteristics of the patients included age, Eastern Cooperative Oncology Group (ECOG) performance status, the presence of diabetes mellitus (DM), and major cardiopulmonary comorbidities. Intraoperative observations were recorded in detail, including the presence of peritoneal carcinomatosis, diaphragmatic disease, small bowel serosal/mesenteric and colon serosal/mesenteric involvement (absent, localised, diffuse), ascites, pleural effusion, omental cake, presence of splenic, and liver metastases (absent, superficial, parenchymal). The presence of extraabdominal lymph nodes was noted by preoperative imaging methods. In light of the findings of randomised controlled trials [21,22], which provide guidance on the management of patients with ovarian cancer, those who were selected for interval debulking were transferred to the medical oncology clinic to receive neoadjuvant chemotherapy (NACT) and underwent surgery approximately 21 days after the last chemotherapy cycle. The number of chemotherapy cycles (3, 4, or ≥6 cycles) was recorded. Surgical resections were recorded as pelvic/paracolic/diaphragmatic peritonectomy (partial, total), small/large bowel resection anastomosis requirement, appendectomy, splenectomy and/or distal pancreatectomy, and lymphadenectomy (selective, systematic). Intraoperative blood transfusion requirement, cytoreduction classification after surgical resection (maximal, optimal, suboptimal), operation time, postoperative intensive care requirement, and hospital stay were recorded. The biochemical laboratory parameters, including CA-125 and albumin levels before debulking surgery, albumin levels on postoperative day 1, and CRP values on the first seven postoperative days, were recorded. FIGO stage and histological subtype of ovarian cancer were recorded as pathological data [23] (Table 1 and Table 2).

### Data Handling and Machine Learning Analysis

To address the issue of missing data in our study, an initial assessment was made to determine the proportion of missing data. The features were then filled using the mean, median or mode, which were missing at low rates. In more complex cases, statistical techniques such as k-nearest neighbour (KNN) imputation and multiple imputation were used. The imputation methods were used to minimise the loss of information in the dataset. In the feature selection stage, we applied a number of methods to improve the performance of the model and eliminate redundant features. The most significant features were identified using Recursive Feature Elimination (RFE), LASSO regularisation and tree-based methods (e.g., Random Forest feature importance). This process improved the efficiency of the model and allowed it to generalise more effectively. To address the high dimensionality of our dataset, we utilised PCA, making use of dimensionality reduction techniques. The application of PCA resulted in a reduction in dimensions while maintaining the underlying variability of the dataset. This approach was advantageous in terms of reducing the risk of overfitting while increasing the effectiveness of the model. By following these steps, we created a reliable, robust, and reproducible method for predicting postoperative complications.

The data were split into 75% and 25% randomly. In total, 134 patient data were used for the training set and 45 patients for the test set. In our study, the data used for the test set (n = 45, 25%) was not included in the algorithm. The algorithm was developed internally with data allocated entirely for the training set. After splitting the data randomly, we made sure to ensure and cross-check that the training and test datasets showed a homogeneous distribution pattern for the subgroups (groups 1 and 2). In this regard, we examined the difference in distribution rates using the z score and found no significant distributional differences between the subgroups in the split data (Table 3).

ML study was carried out with 11 algorithm models, which are the most popular and suitable for our study model, and their prediction capabilities on the test set were noted. The success rates of all algorithms were examined in detail and their prediction capabilities and classification abilities were noted. Machine Learning (ML) models and data mining were evaluated and carried out with WEKA software, version 3.8.6. The models were executed on a Windows 11 operating system utilising an Intel i7 CPU, 16 GB of RAM (Intel, Santa Clara, CA, USA), and NVIDIA GTX 1660 Ti 8 GB graphics card (NVIDIA Corporation, Santa Clara, CA, USA). Mean absolute error, root mean squared error, correlation coefficients, Mathew’s correlation coefficient, and F1 score were used to determine the best performing algorithm. Cohens’s kappa were assessed to analyse model consistency with actual data.

After selecting the most stable and accurate algorithm among the available options, the prediction groups for 45 patients were recorded. The relationship between these predicted values and the actual values was then summarised using a confusion matrix. True positive (TP) rate, false positive (FP) rate, precision, recall, and area under curve (AUC) values were evaluated to demonstrate clinical usability and classification skills.

## 3. Statistical Methods

Statistical analyses in the study were performed using SPSS 27.0 (IBM Inc., Chicago, IL, USA) software. The Kolmogrov–Smirnov test, histogram analysis, skewness/kurtosis data, and Q-Q plots were used to evaluate the assumptions of normal distribution. Proportions within independent groups were investigated with z scores. The qualitative parameters are expressed as frequency and percentage (%). Actual and predicted values were evaluated with crosstabulation matrix. In the entire study, the type-I error rate was taken as 5% (α = 0.05) and *p* < 0.05 level was considered as statistically significant.

## 4. Results

During the study period, there were 179 patients undergoing epithelial ovarian cancer surgery. (Figure 1). A total of 176 ovarian cancer patients underwent surgical procedures for epithelial ovarian cancer. The results showed that 139 patients (77.65%) showed no morbidity or grade I-II CDC morbidity, while 40 patients (22.35%) experienced grade III or higher CDC morbidity.

Table 1 presents the clinical and demographic characteristics of the patients. The mean age was 61.47 ± 12.13 years. Of the patients, 66 (36.87%) had stage III disease and 137 (76.54%) had high-grade histology. A total of 116 patients (64.8%) underwent a primary debulking procedure, while 63 patients (35.2%) underwent an interval debulking procedure following neoadjuvant chemotherapy. The highest CRP values were observed in the first postoperative week, with a median value of 218 on the third day. The median operation time was 300 min, the median intensive care unit stay was 1 day, and the median hospital stay was 11 days. In the first postoperative week, the highest level of CRP values was observed on day 3, with a median value of 218 (Table 2).

Eleven models were evaluated, and it was found that BayesNet was the most effective prediction model. No dominant parameter was observed in the Bayesian net importance matrix plot, as all parameters were taken into account while making predictions. TP rate was 76%, the FP rate was 15.6%, the recall rate (sensitivity) was 76.9%, and the overall accuracy was 82.2% (Table 4). ROC analysis was conducted to predict the CDC grade ≥ III. AUC was found to be 0.863 with a statistical significance of *p* < 0.001 (Figure 1), indicating a high degree of accuracy.

## 5. Discussion

### 5.1. Summary of Main Results

To our knowledge, this is the first study to demonstrate the power of machine learning techniques to predict CDC grade ≥ III complications in patients undergoing surgery for epithelial ovarian cancer. We performed a comparative evaluation of 11 popular models derived from supervised learning and machine learning techniques. The findings revealed that the BayesNet model exhibited the highest degree of efficiency in prediction with a TP rate of 76%, FP rate of 15.6%, overall accuracy of 82.2% and AUC of 0.863 (*p* < 0.001). The Bayesian model identified 10 of 13 patients with grade ≥ III morbidity from a total of 45 patients in the test set.

### 5.2. Conclusions in the Context of Published Literature

Individualised surgical approaches are a common practice in the surgical management of ovarian cancer. Nevertheless, the identification of patients at high-risk of postoperative complications necessitates the utilisation of an accurate and efficacious risk stratification prediction tool [24]. In this retrospective cohort study, we developed and validated machine learning algorithms using 49 preoperative, intraoperative, and follow-up features variables to predict CDC grade ≥ III complications. Many of these features were related to intraoperative observation and surgical resection, indicating the important impact of disease extent and surgical resection on CDC grade ≥ III complications after ovarian cancer surgery.

The incidence of grade ≥ III complications after cytoreductive ovarian cancer surgery is between 10% and 25% [9,25,26,27,28,29]. In our study, this rate was similar to the literature with 22.35%. Hernandez et al. [30] used a machine learning approach to develop a prediction model for CDC grade ≥ III complications after major cancer surgery. The model consisted of 66 features, including demographic information, diagnosis codes, cancer treatment, previous hospitalisations and outpatient visits, medication administration records, oxygen support, vital signs and laboratory tests. Of these, 788 (80%) were used to train and 200 (20%) were reserved to test. The model performed successfully with an AUC of 0.73, sensitivity of 0.78, specificity of 0.63 and accuracy of 0.67 [30]. MySurgeryRisk, a machine learning based risk scoring algorithm for eight major postoperative complications after major surgery, gives successful results with AUC values ranging from 0.82 to 0.94 [14]. Currently, this algorithm is a mobile application where data are manually entered and data are obtained from all systems based on major surgery. Given the unique characteristics and approaches associated with ovarian cancer surgery, we believe that algorithms incorporating disease-specific parameters may be of greater benefit in clinical practice. In a study using 147 variables to predict CDC grade ≥ III complications using patient data from tertiary hospitals specialising in cytoreduction and included in the USA Hyperthermic Intraperitoneal Chemotherapy Collaborative Database, the gradient boosting model outperformed conventional logistic regression models with an AUC of 0.74 [31]. The majority of the database in this study consisted of gastrointestinal cancers; 174 (7.3%) of the total 2372 patients had peritoneal cancer. However, the results of this study are similar to those of our study.

Ovarian cancer outcome prediction, particularly CDC grading, is inherently complex due to the interaction of numerous factors (e.g., tumour stage, patient characteristics, treatment response). BayesNet excels at representing and reasoning about these probabilistic relationships. By learning the dependencies between variables, it can effectively capture the uncertainty associated with each predictor and propagate these uncertainties through the network to arrive at a more robust prediction. BayesNet can handle missing data in a natural way without requiring imputation as some other models do. This can be advantageous when dealing with real-world clinical datasets where missing values are common. On the other hand, BayesNet is effective on sparse data. For models and sample size similar to ours, BayesNet’s ability to learn a simpler structure may be beneficial compared to more complex models prone to overfitting. Random Forests are powerful ensemble methods, but they can sometimes be ‘black boxes’ and lack the interpretability of BayesNet. They can also be prone to over-fitting if the data are limited or noisy. SMO can be highly effective for classification but requires careful tuning of parameters and may be less robust to outliers or missing data than BayesNet. Neural networks excel at learning very complex and non-linear relationships at the expense of a lot of data, the efficiency of BayesNet in dealing with smaller datasets may have led it to outperform MLP in this experiment [32].

### 5.3. Strengths and Weaknesses

Although the database was prospectively maintained, the retrospective nature of the study is the primary limitation. The limited number of patients included in the test set is a secondary limitation, although machine learning algorithms do not lose predictive power like commonly used statistical analyses. The strengths of our study are that it is the first study to be conducted in a group of patients undergoing surgery for epithelial ovarian cancer and that the parameters used in the development of the algorithm are detailed.

## 6. Implications for Practice and Future Research

Identifying patients at high risk of complications in the preoperative period allows a more balanced assessment of the potential risks and benefits of debulking surgery. Timely initiation of postoperative adjuvant chemotherapy is important in this population. The management of serious complications that may occur during this period can be time consuming and this may lead to a delay in adjuvant treatment, which may adversely affect the prognosis of the disease [33]. Although a number of patient- and tumour-related factors that can be determined before surgery are beyond the surgeon’s control, many steps can be taken to reduce the risk of complications. In this way, both surgeons and patients can evaluate potential risks with more concrete data and make informed decisions accordingly. This objective approach not only improves patient safety but also helps develop strategies to prevent potential complications in the postoperative period. The potential of machine learning has been demonstrated in small patient groups. This will likely encourage further studies aimed at predicting outcomes in specific small patient groups.

## 7. Conclusions

The findings of this study showed that the Bayes Net model has a better prediction performance in patients undergoing ovarian cancer surgery compared to other popular machine learning models. Furthermore, our findings demonstrated that machine learning may successfully be utilised in predicting the CDC grade for patients who are receiving ovarian cancer surgery. This knowledge can be standardised in future studies in line with our clinical integration pathway and machine learning methods that can be used in an application or web tool can provide support in terms of risk assessment, decision support systems, management of hospital resources, development of preventive measures, and informing patients, and can benefit both in reducing costs and improving the quality of healthcare services. When the machine learning methods that may eventually be used become available in clinical practice, guidelines will be developed to ensure that ethical standards are met. These guidelines will be developed in collaboration with ethics experts, legal experts and patient advocacy groups.

## Figures and Tables

**Figure 1 medicina-61-00695-f001:**
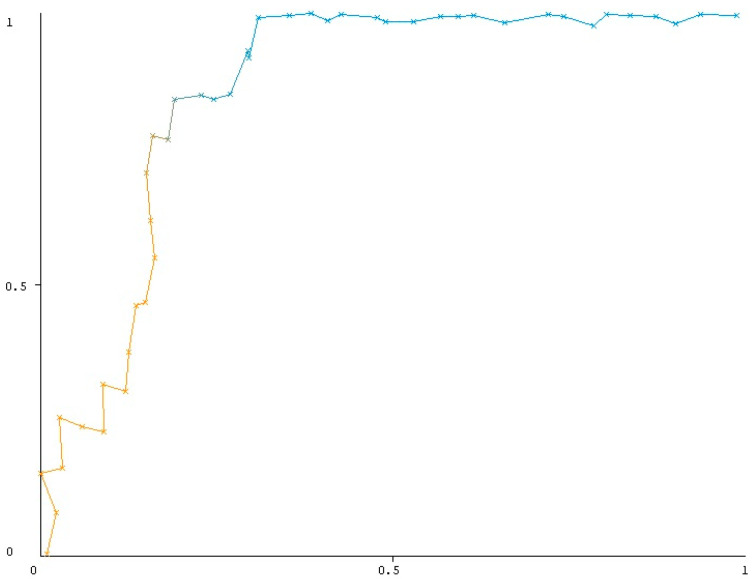
ROC analysis of BayesNet algorithm in predicting CDC grade 3-4-5; (*p* value < 0.001, AUC = 0.863, TP rate = 0.769). Yellow line represents lower thresholds where the classifier tends to classify more instances as positive, leading to a higher True Positive Rate (TPR) but also a higher False Positive Rate (FPR). On the other hand the blue line represents higher thresholds where the classifier becomes more selective in predicting positive instances, leading to a lower FPR. The relevant coloring of the lines was done to emphasize the basic principle of ROC analysis.

**Table 1 medicina-61-00695-t001:** Overall distribution of the nominal parameters used in the ML models.

Parameter		N (%)	Parameter		N (%)	Parameter		N (%)
ECOG PS	0–1	139 (77.65%)	Small bowel mesentery involvemenT	No	119 (66.85%)	Pelvic peritonectomy	No	84 (46.93%)
	≥2	40 (22.35%)		Localised foci	22 (12.36%)		Yes	95 (53.07%)
Major Cardiac Comorbidity	No	135 (75.42%)		Diffuse, military	37 (20.79%)	Paracolic peritonectomy	No	123 (68.72%)
	Yes	44 (24.58%)	Large bowel serosal involvement	No	103 (57.54%)		Yes	56 (31.28%)
Major Pulmonary Comorbidity	No	164 (91.62%)		Localised foci	44 (24.58%)	Peritonectomy (partial or total)	No	81 (45.25%)
	Yes	15 (8.38%)		Diffuse, military	32 (17.88%)		Yes	98 (54.75%)
Diabetes mellitus	No	149 (83.24%)	Large bowel mesentery involvement	No	81 (45.25%)	Diaphragm peritonectomy	No	150 (83.8%)
	Yes	30 (16.76%)		Localised foci	58 (32.4%)		Yes	29 (16.2%)
Neoadjuvant chemotherapy	No	116 (64.8%)		Diffuse, military	40 (22.35%)	Appendectomy	No	111 (62.01%)
	3 cycles	25 (13.97%)	Spleen metastasis	No	159 (88.83%)		Yes	68 (37.99%)
	4 cycles	25 (13.97%)		Yes	20 (11.17%)	Splenectomy and/or distal pancreatectomy	No	156 (87.15%)
	≥6 cycles	13 (7.26%)	Liver metastasis	No	161 (89.94%)		Yes	23 (12.85%)
Debulking surgery	Primary	116 (64.8%)		Any surfacce lesion	12 (6.7%)	Lymphadenectomy	No	59 (32.96%)
	Interval	63 (35.2%)		Parencyhmal	6 (3.35%)		Selective LN debulking	5 (2.79%)
Ascites	No	113 (63.13%)	Pleural effusion	No	140 (78.21%)		Systemic	115 (64.25%)
	Small volume	31 (17.32%)		Yes	39 (21.79%)	Intraoperative need for blood transfusion	No	83 (46.37%)
	Large volume	35 (19.55%)	Extraabdominal LN (+)	No	147 (82.58%)		Yes	96 (53.63%)
Omental cake	No	113 (63.13%)		Yes	31 (17.42%)	Need for intensive care unit	No	72 (40.22%)
	Yes	66 (36.87%)	Cytoreduction	Maximal (no visible)	128 (71.51%)		Yes	107 (59.78%)
Peritoneal carcinomatosis	No	72 (40.22%)		Optimal (<1 cm)	38 (21.23%)	Tumour histotype	High grade	137 (76.54%)
	Localised foci	36 (20.11%)		Suboptimal (≥1 cm)	13 (7.26%)		Others	42 (23.46%)
	Diffuse, military	71 (39.66%)	Intestinal resection	No	138 (77.09%)	FIGO Stage	I	34 (18.99%)
Diaphragmatic disease	No	130 (72.63%)		Yes	41 (22.91%)		II	17 (9.5%)
	Localised foci	12 (6.7%)	Small bowel resection	No	170 (94.97%)		III	66 (36.87%)
	Diffuse, military	37 (20.67%)		Yes	9 (5.03%)		IV	62 (34.64%)
Small bowel serosal involvement	No	132 (74.16%)	Colorectal anastomosis	No	148 (82.68%)	Lymph node involvement	No	75 (41.9%)
	Localised foci	22 (12.36%)		Yes	31 (17.32%)		Yes	43 (24.02%)
	Diffuse, military	24 (13.48%)	İleocolic anastomosis	No	174 (97.21%)		Unknown or no LND	61 (34.08%)
				Yes	5 (2.79%)			

ECOG PS: Eastern Cooperative Oncology Group Performance Status.

**Table 2 medicina-61-00695-t002:** General distribution patterns of the quantitative attributes used in the ML models.

Parameter	Min	Max	Median	Mean	SD
Age	22	82	57	58	11
Serum Ca-125 level prior to debulking surgery	3	19,574	119	604	1913
Serum Albumin level prior to debulking surgery	2	5	4	4	1
Operative time, minutes	120	610	300	322	97
Length of intensive care unit stay, days	0	69	1	3	8
Postoperative length of hospital stay, days	1	77	11	14	9
Day 1, albumin, g/dL	1	43	3	13	13
Day 1, CRP, mg/L	1	317	63	99	85
Day 2, CRP, mg/L	13	435	210	209	91
Day 3, CRP, mg/L	30	499	218	219	95
Day 4, CRP, mg/L	21	443	160	181	101
Day 5, CRP, mg/L	16	417	102	128	92
Day 6, CRP, mg/L	8	455	68	98	84
Day 7, CRP, mg/L	0	439	71	90	75

Min = minimum; max = maximum; SD = Standard deviation.

**Table 3 medicina-61-00695-t003:** The 75% split pattern and the distribution of CD subgroups, and comparison of group 2 proportions within test and training data.

	Training Set	Test Set	Overall
Group 1	107 (79.85%)	32 (71.11%)	139 (77.65%)
Group 2	27 (20.15%)	13 (28.89%)	40 (22.35%)
Overall	134 (74.86%)	45 (25.14%)	179 (100.0%)
z = −1.218 *p* * = 0.223			

Group 1 = no grade, CDC grade 1 or 2. Group 2 = CDC grade III–IV or V. Comparison of the proportions of Group 2 within the training and test sets; indicating a homogeneous distribution. *: Two-tailed comparison of the proportions.

**Table 4 medicina-61-00695-t004:** Evaluation of ML success and prediction capabilities.

Evaluation of Model Success												
	Model				Statistics—Predicting the Desired Group (CDC Grade 3-4-5)							
ML Algorithms *	Time Taken to Build Model	Mean Absolute Error	Root Mean Squared Error	Kappa (κ)	TP Rate	FP Rate	Precision	Recall	Overall Accuracy ^¥^	F1 Score	MCC	ROC (AUC)
J48	<0.001 s	0.2373	0.3849	0.498	0.462	0.031	0.857	0.462	0.822	0.600	0.538	0.776
Logistic	0.08 s	0.3555	0.5963	0.134	0.385	0.250	0.385	0.385	0.644	0.385	0.135	0.669
Simple Logistic ^Φ^	0.22 s	0.2488	0.3526	0.642	0.615	0.031	0.889	0.615	0.867	0.727	0.662	0.880
Random Forest	0.09 s	0.2703	0.3762	0.299	0.231	0.0	1.00	0.231	0.778	0.375	0.419	0.885
BayesNet ^†^	0.03 s	0.2066	0.4143	0.586	0.769	0.156	0.667	0.769	0.822	0.714	0.589	0.863
SMO (Support Vector Machine)	0.03 s	0.3333	0.5774	0.129	0.308	0.188	0.400	0.308	0.667	0.348	0.131	0.560
MultilayerPerceptron	0.03 s	0.3144	0.5331	0.084	0.231	0.156	0.375	0.231	0.667	0.286	0.088	0.690
Random Tree	0.03 s	0.3061	0.5018	0.073	0.154	0.094	0.400	0.154	0.689	0.222	0.087	0.606
Decision Stump	0.02 s	0.2866	0.4023	0.4774	0.538	0.094	0.700	0.538	0.800	0.609	0.485	0.722
Rep Tree	0.03 s	0.2411	0.3716	0.572	0.538	0.031	0.875	0.538	0.844	0.667	0.601	0.739
PART decision	0.03 s	0.2091	0.3888	0.5673	0.692	0.125	0.692	0.692	0.822	0.692	0.567	0.810

Abbreviations: TP = True positive; FP = False positive; MCC = Matthew’s correlation coefficient; ROC = Receiver operating characteristic; AUC = Area under curve; CDC = Clavien–Dingo Classification; s = second. * Training and validation split: 75.0%. ^¥^ In predicting both groups (overall). ^†^ The best performing algorithm in predicting CDC G3-4-5; classified 10 patient (out of 13) as CDC Grade 3-4-5. ^Φ^ The best performing algorithm in general (overall) prediction; classified 8 patients (out of 13) as CDC Grade 3-4-5.

## Data Availability

The data generated in the present study may be requested from the corresponding author.

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
