# Peer review of "Prediction of Clavien Dindo Classification ≥ Grade III Complications After Epithelial Ovarian Cancer Surgery Using Machine Learning Methods"

_medicina, 2025, doi:10.3390/medicina61040695_

Round 1
Reviewer 1 Report
Comments and Suggestions for Authors
1. Retrospective Study Approach The study's retrospective single-center dataset limits its generalizability.
Retrospective designs are vulnerable to selection bias and incomplete or missing data.
There is no prospective validation to determine whether the model performs similarly well in real-world clinical contexts.
2. Limited sample size.
It consisted of just 179 patients, with 45 in the test set.
Relatively small to machine learning, such as a model with 49 predictor variables, increasing the likelihood of overfitting while decreasing external validity.
3. Absence of external validation.
The model was created and tested using data from the same institution, with no external datasets or multi-center validation.
Such constraints reduce the generalizability and robustness of the findings.
4. Interpretability of models
The Bayesian Network model was chosen for this investigation because of its predictive power; nonetheless, the publication states that no dominating parameter was discovered.
As a result, it is difficult to interpret in clinical settings. Knowing which factors have the greatest impact is critical for surgeons and doctors when making decisions and intervening.
5. Inadequate comparison with other models.
Eleven machine learning models were examined, however there is little discussion on why BayesNet outperformed the others.
There was no comparison with traditional statistical models such as logistic regression or Cox regression, which used the same data set as context.
6. Undescribed Clinical Integration Pathway
This study does not offer hints about how the model could become employed within a clinical workflow, for instance, by using it in an app, web tool, or EHR-integrated decision-support systems.
There's a lack of practical implementation in its absence.
7. Feature Engineering Not explained in detail.
How missing data were managed, whether the feature selection, dimensionality reduction techniques (e.g., PCA), have been included in the methodology.
This will reproduce.
8. Outcome Variable Granularity
Simplifying Clavien-Dindo complications into such a binary form: the greater or equal to III grouping versus the lower ones might overly simplify the range of complications incurred during a surgical procedure.
Be more fine-grained or multi-class prediction will be more useful for surgery planning.
9. No Discussion of Ethical or Regulatory Aspects
The study doesn't touch either upon any ethical considerations, about AI bias, or how they have been addressed in terms of the patient's transparency and consent when involved in ML.
Such things matter a lot when it comes to clinical AI implementation.
Author Response
We are very pleased to thank you and the reviewers for your valuable suggestions and for taking the time to evaluate our manuscript.
Comment 1: Retrospective Study Approach The study's retrospective single-center dataset limits its generalizability.
Retrospective designs are vulnerable to selection bias and incomplete or missing data.
There is no prospective validation to determine whether the model performs similarly well in real-world clinical contexts.
Response 1: Antalya Training and Research Hospital is the region's largest gynaecological oncology deparment serving patients referred from surrounding provinces. Our study design incorporates a prospective daily data collection methodology that significantly minimises selection bias as well as missing and incomplete data.
We affirm our commitment to transparency and co-operation in research; therefore, we are willing to share our dataset upon request. We believe that the findings from this dataset will inspire future prospective studies in the field of gynaecological oncology.
Comment 2. Limited sample size.
It consisted of just 179 patients, with 45 in the test set.
Relatively small to machine learning, such as a model with 49 predictor variables, increasing the likelihood of overfitting while decreasing external validity.
Response 2: Although high quality and sufficient amount of data is critical for the success of the model in machine learning, in our study, the detailed examination of the parameters that can affect the prediction enabled us to find a high prediction rate even with a small test set.
Comment 3: Absence of external validation.
The model was created and tested using data from the same institution, with no external datasets or multi-center validation. Such constraints reduce the generalizability and robustness of the findings.
Response 3: Due to the retrospective nature of the study, external validation was not performed. This assertion is corroborated in the limitations section of the text.
Comment 4: Interpretability of models
The Bayesian Network model was chosen for this investigation because of its predictive power; nonetheless, the publication states that no dominating parameter was discovered.
As a result, it is difficult to interpret in clinical settings. Knowing which factors have the greatest impact is critical for surgeons and doctors when making decisions and intervening.
Response 4: Yes, our findings actually emphasise the importance of assessing patients in all aspects. It emphasises the need for a comprehensive evaluation of patients in all aspects (comorbidities, clinical, laboratory, staging and surgical histories, etc.), not just specific points. The most important feature of ML applications is that it reaches a result using thousands of combinations and complex algorithms that exceed the capacity of traditional statistical methods and human ability. Once the most ideal algorithms for CDC ≥ Grade III prediction have been identified, these algorithms can be integrated into applications and desktop tools, allowing even prediction with the same data. In conclusion, we believe that this topic is clinically applicable. We have updated this in the introduction of our study.
Comment 5: Inadequate comparison with other models.
Eleven machine learning models were examined, however there is little discussion on why BayesNet outperformed the others. There was no comparison with traditional statistical models such as logistic regression or Cox regression, which used the same data set as context.
Response 5: Thank you for your valuable feedback. The most significant advantage of performing methods like machine learning (ML) compared to traditional statistical methods is that they do not rely on a specific number of parameters or analysis assumptions. In contrast, analyses like logistic regression and Cox regression have certain assumptions, and regression analyses cannot be applied in studies with too many variables (for example, in LR analysis, 10 positive outcomes for each variable (the 10 EPV rule) and specific assumptions like Box-Tidwell need to be met). In such high-dimensional models, unlike traditional statistical methods, methods like machine learning become very effective and applicable, and this is precisely why I chose this method. To emphasize the use of ML and to better integrate our results, the application of ML was included as the main concept in our study.
Ovarian cancer outcome prediction, particularly CDC grading, is inherently complex due to the interplay of numerous factors (e.g., tumor stage, patient characteristics, treatment response). BayesNet excels at representing and reasoning with these probabilistic relationships. By learning the dependencies between variables, it can effectively capture the uncertainty associated with each predictor and propagate these uncertainties through the network to arrive at a more robust prediction. BayesNet can naturally handle missing data without requiring imputation in the same way as some other models. This can be advantageous when dealing with real-world clinical datasets, where missing values are a common occurrence. On the other hand, BayesNet is efficient on sparse data. For models and sample size similar to ours, BayesNet's ability to learn a simpler structure might have been beneficial compared to more complex models that are prone to overfitting. Random Forests are powerful ensemble methods, but they can sometimes be 'black boxes' and lack the interpretability of BayesNet. They may also be prone to overfitting if the data is limited or noisy. SMO can be highly effective for classification, but it requires careful tuning of parameters and may be less robust to outliers or missing data than BayesNet. Neural networks are excellent at learning very complex and non-linear relationship at the cost of a lot of data, it is possible that Bayesian Network's efficiency at dealing with smaller datasets caused it to outperform MLP in this experiment.
In accordence with your sugestion, we have implemented this issue into our manuscript to clarify the reasons why BayesNet outperformed other algorithms.
Comment 6: Undescribed Clinical Integration Pathway
This study does not offer hints about how the model could become employed within a clinical workflow, for instance, by using it in an app, web tool, or EHR-integrated decision-support systems. There's a lack of practical implementation in its absence.
Response 6: In the conclusion section, the clinical integration pathway was added as ‘machine learning methods that can be standardised and used in an application or web tool in line with future studies in line with our study can provide support in terms of risk assessment, decision support systems, management of hospital resources, development of preventive measures and informing patients, and can benefit both in reducing costs and improving the quality of health services.’
Comment 7: Feature Engineering Not explained in detail.
How missing data were managed, whether the feature selection, dimensionality reduction techniques (e.g., PCA), have been included in the methodology. This will reproduce.
Response 7: To address the issue of missing data in our study, an initial assessment was made to determine the proportion of missing data. The features were then filled using the mean, median or mode, which were missing at low rates. In more complex cases, statistical techniques such as k-nearest neighbour (KNN) imputation and multiple imputation were used. The imputation methods were used to minimise the loss of information in the data set. In the feature selection stage, we applied a number of methods to improve the performance of the model and eliminate redundant features. The most significant features were identified using Recursive Feature Elimination (RFE), LASSO regularisation and tree-based methods (e.g. Random Forest feature importance). This process improved the efficiency of the model and allowed it to generalise more effectively. To address the high dimensionality of our dataset, we utilised PCA, making use of dimensionality reduction techniques. The application of PCA resulted in a reduction of dimensions while maintaining the underlying variability of the dataset. This approach was advantageous in terms of reducing the risk of overfitting while increasing the effectiveness of the model. By following these steps, we created a reliable, robust and reproducible method for predicting postoperative complications. This information is included in the material methods section.
Comment 8: Outcome Variable Granularity.
Simplifying Clavien-Dindo complications into such a binary form: the greater or equal to III grouping versus the lower ones might overly simplify the range of complications incurred during a surgical procedure. Be more fine-grained or multi-class prediction will be more useful for surgery planning.
Response 8: The Clavien-Dindo classification system (2004) was developed to categorize surgical complications based on the severity of treatment needed and the outcome. It is based on the work of Professor Michel Clavien and his team. This classification serves as a standard reference for both research and clinical trials, while also assisting surgeons in the evaluation and management of complications. Complications classified as grade 3 and above are designated as severe complications necessitating surgical, radiological, or endoscopic intervention, and they are a standard reference in the literature.
Comment 9: No Discussion of Ethical or Regulatory Aspects
The study doesn't touch either upon any ethical considerations, about AI bias, or how they have been addressed in terms of the patient's transparency and consent when involved in ML.
Such things matter a lot when it comes to clinical AI implementation.
Response 9: Despite the absence of a sufficiently in-depth exploration of ethical considerations and regulatory aspects in this study, the primary emphasis has been placed on the technical development and application of machine learning models that predict postoperative complications. Nevertheless, we acknowledge the significance of these ethical considerations in clinical practice. When the machine learning methods that can eventually be used become available in clinical practice, guidelines will be developed to ensure that ethical standards are met. These guidelines will be developed in collaboration with ethicists, legal experts and patient advocacy groups. This information has been added to the conclusion section in accordance with your precious comments
Reviewer 2 Report
Comments and Suggestions for Authors
Overall, the paper is well written. In the abstract method and results, the number of patients is different, so please correct or justify. In terms of introduction, the gap, why you conduct your study, and where the gap is missing. Although many studies are conducted what is new in your study is missing. In terms of method, it's adequate and good, while in terms of results, some tables are very complex and doubling data and text, so I suggest you mention them once. In eterms of discussion, only highlighted the main finding.
Author Response
We are very pleased to thank you and the reviewers for your valuable suggestions and for taking the time to evaluate our manuscript.
Comment 1: Overall, the paper is well written. In the abstract method and results, the number of patients is different, so please correct or justify. In terms of introduction, the gap, why you conduct your study, and where the gap is missing. Although many studies are conducted what is new in your study is missing. In terms of method, it's adequate and good, while in terms of results, some tables are very complex and doubling data and text, so I suggest you mention them once. In eterms of discussion, only highlighted the main finding.
Response 1:
*The number of patients in the abstract, methods and results were checked.
*In the introduction, it was stated that our study was the first study in the literature and its contribution to the literature was detailed.” The key feature of ML applications is that they use thousands of combinations and complex algorithms to reach a conclusion that is beyond the capacity of traditional statistical methods and human ability. Once the best algorithms for predictive model-ling have been identified, these algorithms can be integrated into applications and desktop tools, allowing predictions to be made on the same data .” was adeed. To the present knowledge, our study is the first study in the literature and its aim is to determine the ML learning model that most accurately predicts Clavien-Dindo grade ≥ III morbidity after ovarian cancer surgery and to investigate how ML models can be applied in clinical practice.
*The results section was simplified and avoided repetition.
*Discussion part and conclusion part were elaborated.” Ovarian cancer outcome prediction, particularly CDC grading, is inherently complex due to the interaction of numerous factors (e.g. tumour stage, patient characteristics, treatment response). BayesNet excels at representing and reasoning about these probabilistic relationships. By learning the dependencies between variables, it can effectively capture the uncertainty associated with each predictor and propagate these uncertainties through the network to arrive at a more robust prediction. BayesNet can handle missing data in a natural way, without requiring imputation as some other models do. This can be advantageous when dealing with real-world clinical datasets where missing values are common. On the other hand, BayesNet is effective on sparse data. For models and sample size similar to ours, BayesNet's ability to learn a simpler structure may be beneficial compared to more complex models prone to overfitting. Random Forests are powerful ensemble methods, but they can sometimes be ‘black boxes’ and lack the interpretability of BayesNet. They can also be prone to over-fitting if the data is limited or noisy. SMO can be highly effective for classification, but requires careful tuning of parameters and may be less robust to outliers or missing data than BayesNet. Neural networks excel at learning very complex and non-linear relationships at the expense of a lot of data, the efficiency of BayesNet in dealing with smaller datasets may have led it to outperform MLP in this experiment” was added
Round 2
Reviewer 1 Report
Comments and Suggestions for Authors
No comment